# Newborn screening timeliness quality improvement initiative: Impact of national recommendations and data repository

Marci K. Sontag[1,2]*, Joshua I. Miller[1,2], Sarah McKasson[1,2], Ruthanne Sheller[3], Sari Edelman[3], Careema Yusuf[3], Sikha Singh[3], Deboshree Sarkar[4], Joseph Bocchini[5], Joan Scott[4], Jelili Ojodu[3], Yvonne Kellar-Guenther[1,6]

1 Center for Public Health Innovation, CI International, Littleton, Colorado, United States of America, 2 Department of Epidemiology, Colorado School of Public Health, University of Colorado Denver, Aurora, Colorado, United States of America, 3 Association of Public Health Laboratories, Silver Spring, Maryland, United States of America, 4 Maternal and Child Health Bureau, Health Resources and Services Administration, U.S. Department of Health and Human Services Rockville, Rockville, Maryland, United States of America, 5 Department of Pediatrics, Willis-Knighton Health System, Tulane University School of Medicine, Shreveport, Louisiana, United States of America, 6 Department of Community and Behavioral Health, Colorado School of Public Health, University of Colorado Denver, Aurora, Colorado, United States of America

* msontag@ciinternational.com

**Data Availability Statement:** Data cannot be shared publicly because of signed memoranda of understanding between state newborn screening

## Abstract

### Background

Newborn screening (NBS) aims to achieve early identification and treatment of affected infants prior to onset of symptoms. The timely completion of each step (i.e., specimen collection, transport, testing, result reporting), is critical for early diagnosis. Goals developed by the Secretary of Health and Human Services' Advisory Committee on Heritable Disorders in Newborns and Children (ACHDNC) for NBS timeliness were adopted (time-critical results reported by five days of life, and non-time-critical results reported by day seven), and implemented into a multi-year quality improvement initiative (NewSTEPS 360) aimed to decrease the time to result reporting and intervention.

### Methods

The NBS system from specimen collection through reporting of results was assessed (bloodspot specimen collection, specimen shipping, sample testing, and result reporting). Annual data from 25 participating NBS programs were analyzed; the medians (and interquartile range, IQR) of state-specific percent of specimens that met the goal are presented.

### Results

The percent of specimens collected before 48 hours of life increased from 95% (88–97%) in 2016 to 97% (IQR 92–98%) in 2018 for the 25 states, with 20 (80%) of programs collecting more than 90% of the specimens within 48 hours of birth. Approximately 41% (IQR 29–57%) of specimens were transported within one day of collection. Time-critical result reporting in the first five days of life improved from 49% (IQR 26–74%) in 2016 to 64% (42%-71%)

programs and the Association of Public Health Laboratories. Data requests are reviewed by the NewSTEPs Data Review Workgroup (contact via careema.yusuf@aphl.org) for individuals who meet the criteria for access to state-level data.

**Funding:** The project described in this article was funded by the Department of Health and Human Services, Health Resources and Services Administration under Cooperative Agreements #UG8MC28554 [MKS] and #U22MC24078 [JO] and the Cystic Fibrosis Foundation under Grant Number SONTAG16Q10 [MKS]. The HRSA provided support in the form of salaries for authors [MKS, JIM, SM, CY, RS, SE, SS, JO, YKG], but did not have any additional role in the study design, or data collection and analysis. CI international provided support in the form of salaries for some of the authors [MKS, JIM, YKG]. The specific roles of these authors are articulated in the 'author contributions' section. The HSRA did however participate in the decision to publish as well as in the preparation of the manuscript. The CFF and CI International did not have any role in the study design, or data collection and analysis, the decision to publish, or preparation of the manuscript.

**Competing interests:** The authors have read the journal's policy and have the following competing interests: MKS, JIM, and YKG are paid employees of CI International. The Cystic Fibrosis Foundation provided support for this study under Grant Number SONTAG16Q10 [MKS]. This does not alter our adherence to PLOS ONE policies on sharing data and materials. There are no patents, products in development or marketed products to declare.

**Abbreviations:** ACHDNC, Advisory Committee on Heritable Disorders in Newborns and Children; ACMG, American College of Medical Genetics and Genomics; DBS, Dried Blood Spot; HHS, U.S. Department of Health and Human Services; HIT, Health Information Technology; HRSA, Health Resources and Services Administration; MOU, Memorandum of Understanding; NBS, Newborn Screening; NewSTEPs, Newborn Screening Technical assistance and Evaluation Program; QI, Quality Indicator; RUSP, Recommended Uniform Screening Panel.

in 2018, and for non-time critical results from 64% (IQR 58%-78%) in 2016 to 81% (IQR 68–91%) in 2018. Laboratories open seven days a week in 2018 reported 95% of time-critical results within five days, compared to those open six days (62%), and five days (45%).

## Conclusion

NBS programs that participated in NewSTEPs 360 made great strides in improving timeliness; however, ongoing quality improvement efforts are needed in order to ensure all infants receive a timely diagnosis.

## Introduction

### Newborn screening

Newborn screening (NBS) is a public health program that aims to identify newborns at risk for serious life-altering disorders in the first week of life. The NBS process is composed of multiple components (Fig 1) that must work in a coordinated and efficient manner to allow for early medical intervention before significant and irreversible damage occurs.[1] Hospital staff, midwives, and other clinical personnel collect, package, and ship the dried blood spot NBS specimen through commercial or private couriers to be delivered to a state, regional, or private NBS laboratory. Once the specimen is received at the laboratory, testing is completed and results are reported to the appropriate medical personnel who confirm or rule out a diagnosis and initiate the required intervention as appropriate.

### Timeliness in newborn screening

Timely identification of newborns affected by core disorders on the Advisory Committee on Heritable Disorders in Newborns and Children (ACHDNC) Recommended Uniform Screening Panel (RUSP) is critical.[2, 3] The ACHDNC has the mission to reduce morbidity and mortality due to heritable disorders in newborns and children, and provides recommendations to guide and strengthen the newborn screening system. While early detection has always been the goal of NBS, the expansion of the list of screened disorders in the late 1990s and early 2000s to include those identified via tandem-mass spectrometry with a short pre-symptomatic window has led to an increased urgency to detect affected newborns as quickly as possible.[4, 5] NBS timeliness recommendations were first published in 2006 by the American College of Medical Genetics and Genomics (ACMG) and included specifications that all specimens should arrive at the NBS laboratory within three days of collection, and that results be reported within two days of specimen receipt and within five days of specimen collection.[1]

In 2013, based on public comment, the ACHDNC decided to review policies and practices relating to timeliness of NBS in the United States. In support of this work, the Society for Inherited Metabolic Disorders (SIMD) classified 16 of 35 disorders included on the RUSP as time-critical, requiring immediate medical attention. [6] Based on methodologies in practice, published literature and expert opinion, in 2015, the ACHDNC developed five timeframes for conducting newborn screening (Table 1). [7]

### NewSTEPs 360

In February 2015, the Health Resources and Services Administration (HRSA) funded a collaborative improvement and innovation network to support multidisciplinary teams in

## Pre-Analytic Phase

## Analytic → Post-Analytic Phase

**Fig 1. Newborn screening process, illustrating specimen collection through result reporting.** Newborn screening (NBS) is a complex system that involves the collection of specimens at birthing facilities, transportation of specimens to the NBS public health laboratory for testing and communicating results to health care providers and families. Each step needs to occur in a timely manner in order to prevent infant mortality and morbidity. NewSTEPs 360 supported state/territorial NBS programs to address challenges associated with the pre-analytical and analytical phases of the NBS process by implementing various activities, including 1) providing education to birthing centers and midwives about the importance of timely collection and shipment of specimens; 2) shortening transit time by optimizing shipping methods; 3) expanding laboratory operating hours to decrease the time from specimen receipt to results reporting; 4) improving the efficiency of laboratory workflows; and 5) developing a health information technology infrastructure to improve the transmission of electronic demographic information, laboratory orders, and results between the NBS laboratory and health care providers.

**Table 1. Newborn screening timeliness goals from the Advisory Committee on Heritable Disorders in Newborns and Children (ACHDNC) [7].**

| |
|---|
| **To achieve the goals of timely diagnosis and treatment of screened conditions and to avoid associated disability, morbidity and mortality, the following time frames should be achieved by NBS systems for the initial newborn screening specimen:** |
| • **Presumptive positive results for time-critical conditions should be communicated immediately to the newborn's health care provider but no later than five days of life.** |
| • **Presumptive positive results for all other conditions should be communicated to the newborn's health care provider as soon as possible but no later than seven days of life.** |
| • **All NBS tests should be completed within seven days of life, with results reported to the health care provider as soon as possible.** |
| **In order to achieve the above goals:** |
| • **Initial NBS specimens should be collected in the appropriate time frame for the newborn's condition but no later than 48 hours after birth.** |
| • **NBS specimens should be received at the laboratory as soon as possible; ideally within 24 hours of collection.** |

improving newborn screening timeliness. This project was called NewSTEPs 360. Supplemental funding to support this project was also provided by the Cystic Fibrosis Foundation. Under NewSTEPs 360, NBS programs were convened to identify and overcome barriers to timely NBS through technical and financial assistance. NewSTEPs 360 was built upon the foundation of the HRSA-funded Newborn Screening Technical assistance and Evaluation Program (NewSTEPs)[8], which included access to the NewSTEPs Data Repository.

The NewSTEPs Data Repository collects data on NBS system components with the goal of supporting quality improvement initiatives and providing comparative data at the state, regional, and national levels. NBS programs that voluntarily enter data into the repository have access to their own data plus de-identified, aggregate data from other participating programs. Data elements collected in the repository include NBS program information (e.g., disorders screened, fees, policies, program structure, etc.), quality indicators (QI) for each stage of the NBS process at the programmatic level, and case data [9] on infants with a confirmed diagnosis of a disorder detected by NBS.

NewSTEPs adopted a panel of eight quality indicators that measure newborn screening program performance that were developed by the broader newborn screening community, including newborn screening laboratorians, follow-up specialists, and clinical providers.[10] A subset of the panel of QIs that reflect timeliness outcomes were collected as part of NewSTEPs 360 (Table 2). Each NBS program participating in NewSTEPs 360 was assigned a continuous quality improvement (CQI) coach who met with the NBS program team monthly to identify challenges and opportunities for improvement in this subset of QIs.

This study summarizes the impact of implementing quality improvement efforts in participating NBS programs to attain national timeliness recommendations for newborn screening. We evaluated the timeliness of initial specimen collection, delivery from the birthing center to the NBS laboratory, reporting of results for both time-critical and non-time-critical disorders, and the overall reporting of all NBS results. In a subset of programs, we also assessed the timeliness of diagnosis and medical intervention. Finally, we analyzed the impact of individual program activities to improve timeliness.

**Table 2. NewSTEPs Data Repository quality indicator 5: NBS timeliness from collection to reporting results.**
Excerpted from NBS quality indicator panel. [10].

| Quality Indicator | Description |
|---|---|
| QI 5a.i | Percent of first dried blood spot specimens collected in the specified time intervals, in units of hours, from birth. |
| QI 5b.i | Percent of first dried blood spot specimens received at the NBS laboratory in the specified time intervals, in the unit of days, from specimen collection. |
| QI 5c.i | Percent of dried blood spot specimens with out-of-range results for time-critical disorders reported in the specified time intervals, in units of days, from laboratory receipt. |
| QI 5c.ii | Percent of dried blood spot specimens with out-of-range results for non-time-critical disorders reported in the specified time intervals, in units of days, from laboratory receipt. |
| QI 5c.iii | Percent of first specimens with normal or out-of-range results for any disorder reported in the specified time intervals, in units of days, from laboratory receipt. |
| QI 5d.i | Percent of dried blood spot specimens with out-of-range results for time-critical disorders reported in the specified time intervals, in units of days, from birth. |
| QI 5d.ii | Percent of dried blood spot specimens with out-of-range results for non-time-critical disorders reported in the specified time intervals, in units of days, from birth. |
| QI 5d.iii | Percent of first specimens with normal or out-of-range results for any disorder reported in the specified time intervals, in units of days, from birth. |

## Methods

### Participants in NewSTEPs 360

Twenty-eight US state and territorial NBS programs were selected to participate in NewSTEPs 360 via two rounds of a competitive application process. State newborn screening programs applied through an internet-based application process, identifying the challenges within their programs, and proposed corresponding quality improvement initiatives. Baseline data were required from all applicants. Sixteen applicants representing 20 states were selected for participation in 2015 and a second cohort of eight programs joined in 2016. Participating programs actively engaged in a continuous quality improvement framework to improve timeliness by developing individual improvement projects at the programmatic level. Funding for NewSTEPs 360 was provided by HRSA (UG8MC28554, 9/1/15–8/31/18, no-cost extension through 8/31/19). Supplemental funding was provided by the Cystic Fibrosis Foundation (SON-TAG16Q10). The infrastructure for NewSTEPs is funded through HRSA (U22MC24078). NewSTEPs activities were deemed to be public health quality improvement and not human subject research by the Colorado Multiple Institutional Review Board.

### Data collection

**NewSTEPs data repository and data security.** The repository is a centralized web-based platform that can be accessed by registered users via a 128-bit secure socket layer (SSL) encryption. Registration for the NewSTEPs repository is open to all interested parties, however access to state specific data elements is restricted to individuals working in the state newborn screening system. NewSTEPs requires that NBS programs have a signed Memorandum of Understanding (MOU) with the Association of Public Health Laboratories (APHL) in order to enter Quality Indicator (QI) and case data into the repository. Review of QI and case-level data are limited by role-based access control that was assigned at the individual NBS program level, whereas review of programmatic NBS program information (i.e. operating hours, policies and procedures, state demographics) is available to the public.

**Quality indicator data.** NewSTEPs 360 participants were required to provide monthly data for the QIs associated with timeliness,[10] which was aggregated at the annual level for cross-year comparison. To further encourage data entry and accuracy, CQI coaches frequently reviewed the data using visualization tools and discussed progress or barriers during the monthly or bi-monthly team coaching calls.

**Quality indicator benchmarks.** Benchmarks were adapted directly from the ACHDNC timeliness goal recommendations (https://www.hrsa.gov/advisory-committees/heritable-disorders/newborn-screening-timeliness.html). Few newborn screening programs were able to meet the ambitious recommendation stated by the ACHDNC the NBS specimens should be received at the laboratory as soon as possible, ideally within 24 hours of collection. In response to this, NewSTEPs created an additional benchmark of two calendar days to assess time elapsed from specimen collection to receipt at the laboratory as an intermediary step. For purposes of our analysis, we equated 24-hours to the next calendar day. Assessing timing of specimen receipt per calendar day is in better alignment with the typical newborn screening laboratory approach. Programs typically test specimens after scheduled shipments have arrived on a given day, shipments arriving earlier in the day may not be tested earlier than those arriving right before the scheduled testing time, making calendar day a more meaningful metric than hours of delivery.

Further, an additional metric was added to assess the time from specimen receipt at the laboratory to results being reporting. This added metric helps identify opportunities for

**Table 3. New metric added to calculate time elapsed from specimen receipt at the NBS laboratory to reporting results.**

| | ACHDNC Timeliness Goals | | | New Metrics |
| --- | --- | --- | --- | --- |
| | Birth to Reporting results (A) | Birth to Collection (B) | Specimen Collection to Receipt at Lab (C) | Specimen receipt to reporting (D = A-B-C) |
| **Time-Critical** | 5 Days | 2 Days | 1 Day | 2 Days |
| **Non-Time-Critical** | 7 Days | 2 Days | 1 Day | 4 Days |

improving laboratory processes that could affect overall timeliness. The additional benchmark was calculated based upon the benchmarks set by the ACHDNC for other timeliness outcomes. The calculation of specimen receipt to reporting results are provided in Table 3.

**Case data.** The NewSTEPs Data Repository collects basic demographic and diagnostic information on all newborns with a disorder diagnosed through NBS in the US. Continuous timeliness measures were collected on each confirmed case to understand the factors that lead to shortened intervention times Case data is collected in the year following the birth of an infant to allow for the diagnostic process to be completed; cases entered for 2015–2017 by NewSTEPs 360 programs were included in this analysis.

NewSTEPs has implemented the following definitions for intervention and diagnosis, with disorder-specific definitions available[11]:

- Time to medical intervention: The first time a medical professional acts to change the course of care for an infant. Intervention may occur via phone or clinic visit. This may also include the date a decision was made NOT to change course for the infant.

- Time to diagnosis: The time elapsed from birth until a biochemical or molecular test result on a specimen taken from the infant that confirm the NBS result reported.

**Strategies employed to improve timeliness.** NewSTEPs 360 guided participating NBS programs through CQI activities via training, personalized coaching, and interactive learning sessions between NBS programs. To support team development and growth, a Plan-Do-Study-Act (PDSA) personality tool was developed to support team growth and team members' roles within their programs (S1 File). The tool is a short quiz that helps a team to determine if members naturally affiliate with one functional component of the PDSA cycle to better understand how the team functions together; subsequent discussions led teams to identify strategies to maximize team productivity when engaging the PDSA cycle. Programmatic activities varied (Table 4) from educational strategies for birthing facilities and health care providers to increasing courier services and operating hours (Timeliness Toolkit for Expanding Newborn Screening Services–S2 File) and improving health information technology (HIT) systems (Building Block Guide—S3 File). Successes and failures were shared within the participating programs to facilitate the continuous quality improvement environment.

## Statistical analysis

Monthly quality improvement data reported for January 2016 –December 2018 were converted to annual metrics and were included in the analysis. Timeliness Quality Indicator (QI) data were collected for the purposes of program improvement on a national level and are not intended for formal statistical analysis. Each participating state newborn screening program provides data for this repository with the intent of informing decisions to improve newborn screening systems. NewSTEPs 360 was a quality improvement initiative that was not powered to detect statistical differences. Further, small cell sizes for individual improvement categories

**Table 4. List of strategies participating NewSTEPs 360 programs used to improve NBS timeliness.** * Each Strategy is Linked to a Corresponding QI Solution in Fig 1.

**Specimen Collection (QI Solution 1)**

Educated birthing facility staff and midwives/community-based providers on NBS timeliness (i.e., collection and shipment of blood spot specimens, reducing unsatisfactory specimens, and completion of the NBS card to minimize missing demographic information).

Educated health care providers on the timeliness of follow-up and diagnosis.

**Specimen Shipment (QI Solution 2)**

Educated birthing facility staff and midwives around courier pick-up hours and location.

Educated couriers on the importance of NBS specimens and timeliness of pick-up to drop-off at the NBS laboratory.

Expanded specimen pick-up to include weekends and/or holidays for all or some birthing facilities; utilize label/sticker to indicate weekend or holiday delivery.

Implemented new courier and/or changed courier route to reduce delivery time from pick-up to delivery at the NBS laboratory.

Implemented centralized "drop-off" locations for out-of-hospital births (i.e., FedEx, local/county health departments or neighboring hospitals); provided UPS labels for midwives.

Changed pick-up location in hospitals to reduce "handoffs."

Built-in contract monitoring practice for couriers; establish cut-off times for specimen delivery.

**Specimen Receipt (QI Solution 3)**

Expanded operating hours to include weekends and/or holidays so that the laboratory is not closed for more than two consecutive days.

Modified shift hours so that laboratory staff is available to accession specimens upon delivery; modify cut-off times for specimen delivery to align with hours of operation.

**Lab Testing (QI Solution 4)**

Implemented alternative testing methodologies, workflows, and/or algorithms to improve time from receipt to reporting results.

Improved laboratory information systems to minimize demographic errors and/or link specimens.

Hired quality improvement and/or data analytic staff

**Results Reporting (QI Solution 5)**

Implemented electronic ordering and/or electronic result reporting via HL7 interfaces.

Developed web portal for birthing providers or physicians for more timely access to NBS results.

Implemented electronic mechanisms for demographic data entry (e.g., optical character recognition, printing labels for newborn screening blood spot specimen card).

**Other Strategies**

Changed regulations for blood spot collection to 24 to 48 hours after birth.

*Additional tools available in the NewSTEPs Resource Library [12]

may result in spurious significant results. As a result, descriptive statistics and graphical displays were created, presenting the changes in the percent of specimens meeting ACHDNC benchmarks for time elapsed from birth to specimen collection, collection to laboratory receipt, laboratory receipt to reporting out NBS results, and birth to reporting out NBS results. Additional investigations of the timeliness indicators were completed, stratified by days of operation and type of laboratory (local state laboratory vs. external [regional or private]). Differences in individual case data were tested using non-parametric Wilcoxon-ranked sum tests, and a significance level of 0.05 was set. Data were analyzed using SAS version 9.4 (Cary, NC), and displayed using Tableau Desktop (Seattle, WA, copyright 2019). Results of the analysis do not display state or territory names with the intent to protect NBS programs from the release of sensitive information.

The Colorado Multiple Institutional Review Board determined that the newborn screening quality improvement initiaitives led developed through NewSTEPs are not human subject research.

**Table 5. Percent of newborn screening dried blood spot specimens from each program that met the timeliness goals, by year; medians of the programs' results are reported; N is the number of programs that provided data for the specified quality indicator.** For example: in 2016, half of the programs reported that at least 95.1% of the specimens met the goal of birth to specimen collection.

| QI Timeliness Measure | Time Frame | N | 2016 | | 2017 | | 2018 | |
|---|---|---|---|---|---|---|---|---|
| | | | Median % | IQR | Median % | IQR | Median % | IQR |
| Birth to specimen collection | 48 Hours | 25 | 95.1% | 88.1% - 97.4% | 96.4% | 90.8% - 97.8% | 97.0% | 92.4% - 98.3% |
| Specimen collection to receipt at lab | 1 Day | 19 | 40.0% | 28.6% - 52.5% | 39.4 | 30.4% - 56.4% | 41.8% | 28.9% - 56.5% |
| Specimen collection to receipt at lab | 2 Days | 19 | 74.3% | 67.8% - 86.6% | 79.6% | 69.9% - 88.7% | 80.9% | 70.3% - 88.45% |
| **Receipt to Reporting Results** | | | | | | | | |
| Presumptive positive* for time-critical disorders | 2 Days | 16 | 65.5% | 38.0% - 89.9% | 69.7% | 50.2% - 88.4% | 75.8% | 50.5% - 90.4% |
| Presumptive positive* for non-time-critical disorders | 4 Days | 15 | 80.2% | 56.9%-93.9% | 90.0% | 72.4% -95.1% | 93.5% | 67.3% - 96.3% |
| All (normal and presumptive positive results) | 4 Days | 19 | 90.3% | 69.1% - 98.8% | 90.8% | 83.0% - 99.2% | 94.2% | 88.3% - 99.3% |
| **Birth to Reporting Results** | | | | | | | | |
| Presumptive positive* for time-critical disorders | 5 Days | 16 | 48.9% | 25.8% - 73.8% | 48.8% | 34.3% - 71.5% | 63.5% | 42.5%-71.0% |
| Presumptive positive* for non-time-critical disorders | 7 Days | 15 | 64.4% | 57.8% - 77.9% | 75.9% | 67.1% - 86.0% | 80.9% | 68.0% - 90.7% |
| All (normal and presumptive positive results) | 7 Days | 18 | 88.9% | 68.8% - 96% | 87.6% | 78.4% - 95.7% | 89.5% | 84.8% - 98.2% |

* Presumptive positive indicates with high probability that the infant may have the disorder; however confirmatory diagnostic testing is required.

## Results

### Timeliness quality indicators

Twenty-eight newborn screening programs participated in NewSTEPs 360; 25 provided complete data for birth to specimen collection (2016–2018), and a subset of those programs provided data for each of the other timeliness QIs (Table 5). Participating programs that provided data for the three years are presented, although some programs were not able to provide data for all of the QIs due to systems challenges and laboratory information management systems structure that did not allow for data collection or retrieval at the program level. Progression toward meeting the ACHDNC timeliness goals for all QIs was achieved in most programs, demonstrated through individual trajectories representing the percent of specimens that met the goal (Fig 2A–2C). The median of all programs for each indicator demonstrated improvement in all indicators across all three years, described in detail below.

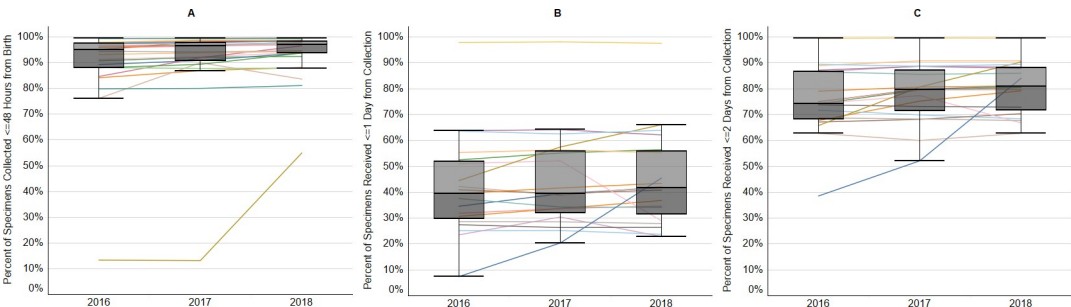

**Fig 2. Percent of newborn screening dried blood spot specimens that achieved timeliness goals for collection and receipt at the testing laboratory.** Data are presented for each state program individually, with box plots overlaid to demonstrate national trends. Box and whisker plots display the median of the percent of specimens that met the benchmark for each program, with upper and lower quartiles, and range. Panel A: Percent of specimens collected within 48 hours of birth, Panel B: Percent of specimens received at the laboratory within one day of collection (next calendar day), Panel C: Percent of specimens received at the laboratory within two calendar days of collection.

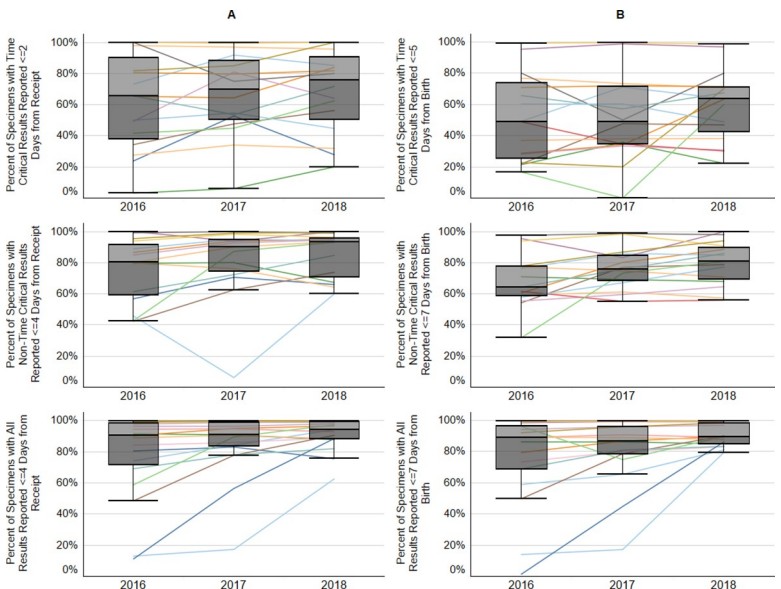

**Fig 3. Percent of newborn screening dried blood spot specimens with results reported within the recommended timeframe.** Data are presented for each state program individually, with box plots overlaid to demonstrate national trends. Box and whisker plots display the median of the percent of specimens that met the benchmark for each program, with upper and lower quartiles, and range. Panel A: Percent of specimens with results reported out for time-critical results within two days of receipt at lab (top), non-time-critical reported out within four days of receipt (middle), and all results reported out within four days of receipt (bottom); Panel B: Percent of specimens with results reported out for time-critical results within five days of birth (top), non-time-critical reported out within seven days of birth (middle), and all results reported out within seven days after birth (bottom).

**Specimen collection and transportation.** Programs successfully collected initial specimens within the first 48 hours of birth, with stepwise increases demonstrated each year (Fig 2A, Table 5; median of the programs' percent of specimens that met benchmark in 2016: 95.1%, 2017: 96.4%, 2018: 97.0%). In 2018, more than 90% of specimens were collected in the 48-hour time frame in 19 programs (n = 19/25, 76%) and more than 95% were collected within 48 hours in 14 programs (n = 14/25, 56%). Specimens deemed to be unsatisfactory for analysis by the NBS laboratory were flagged for recollection as they could result in delayed testing and subsequent reporting of results. The percent of specimens deemed unsatisfactory for analysis by state laboratories was 1.3% in 2016, 1.3% in 2017, and 1.5% in 2018 (medians of all programs). Further, NBS cards that were submitted without complete essential demographic information could have also delayed testing and reporting of results. Variable definitions between NBS programs and changes in the definitions of the required information within state programs made these data difficult to summarize for programmatic trends.

The program median for the time from collection to receipt at the NBS laboratory (Fig 2B, Table 5) on the next calendar day was 40.0% in 2016, 39.4% in 2017 and 41.8% in 2018. Allowing two calendar days after collection to receipt (Fig 2C), the program median of specimens which met the guideline increased to 74.3% in 2016, to 80.9% in 2017 and 81% in 2018.

## Reporting of NBS results

The percentage of specimens with reporting times that met the benchmarks improved both in individual state trajectories and in program medians over all three years (Fig 3A, Table 5). The timely reporting of NBS results for time-critical disorders within two days of receipt at the laboratory improved from a program median of 65.5% in 2016 to 75.8% in 2018. Similarly, the

program median for non-time-critical result reporting within four days of laboratory receipt improved from 80.2% in 2016 to 93.5% in 2018. Reporting of all NBS results from laboratory receipt improved from 90.3% in 2016 to 94.2% in 2018.

The elapsed time from birth to result reporting showed improvements in each category as well (Fig 3B, Table 5). Time-critical results reported within five days of birth started at 48.9% in 2016 and increased to 63.5% by 2018; reporting of non-time-critical results within seven days of birth improved from 64.3% to 80.9%. The program median for reporting all NBS results within seven days of birth did not demonstrate change over this period (88.9% in 2106 to 89.5% in 2018).

**Timeliness data for cases with a confirmed diagnosis.** The 25 participating NBS programs that provided data for NewSTEPs 360 reported 1,713 cases with a confirmed diagnosis of a disorder identified by newborn screening for the years 2016–2018 (288 time-critical cases; 1,425 non-time-critical cases). Individual specimen collection times are consistent with ACHDNC timeliness goals (Table 6). The median report time for time-critical disorders (five days) was earlier (p < 0.0001) than non-time-critical disorders (seven days), and both are in alignment with the timeliness goals. The resulting intervention and diagnosis times are earlier for time-critical disorders than non-time-critical (p<0.0001).

Individual-level data from diagnosed cases demonstrate that at least 50% of the specimens were collected, received, and results were reported within the ACHDNC recommended period. Intervention and diagnosis occur earlier in infants with time-critical disorders compared to infants with non-time-critical disorders, reflecting the expedited nature of laboratory processes within the laboratories related to time-critical disorders.

**Laboratory operating hours.** We found that laboratory operating hours are a critical factor associated with improved specimen delivery times, timely testing, and efficient reporting of results. Each state reported wheteher they were open 5, 6, or 7 days, along with the activites performed on those days. During the NewSTEPs 360 program, two participating NBS programs increased the number of days their laboratories were open, and multiple programs added or adjusted the hours of operation to align with the delivery of samples. By 2018, six of the 25 NBS laboratories were open five days a week, 13 were open six days a week, and six were open all seven days. Activities performed on a given day of the week by laboratories vary (Table 7) due to staff training, availability, and internal policy decisions. For example, programs may report being open 7 dyas a week while not reporting non-time-critical results due to an agreement with clinical specialists to wait until regular business hours to avoid prolonged waiting times for families with a presumptive positive for non-time-critical disorders. These

**Table 6. Timeliness metrics for newborns identified with a disorder on the newborn screening panel.**

|  | All Infants (n = 1,713) median (IQR) | Infants with a Time-Critical Disorder [6] (n = 288) median (IQR) | Infants with a Non-Time-Critical Disorder (n = 1,425) median (IQR) |
|---|---|---|---|
| **Collection of Specimens (hours)** | 28 (24–40) | 28 (25–38) | 28 (24–41) |
| **Receipt at Lab (days from birth)** | 3 (3–4) | 3 (3–4) | 3 (3–4) |
| **Result Release (days from birth)** | 6 (5–8) | 5 (4–7) | 7 (5–8) |
| **Intervention (days from birth)** | 11 (6–26) | 6 (4–9) | 13 (7–29) |
| **Diagnosis (days from birth)** | 18 (9–39) | 12 (7–33) | 20 (10–40) |

**Table 7. Laboratory weekend operations vary across newborn screening programs, categorized by the number of days a laboratory reports testing dried blood slot specimens*.**

| NBS Lab Activity | Laboratory Reported Testing 7 days/week | | | | Laboratory Reported Testing 6 days/ week | | | | Laboratory Reported Testing 5 days/week | | | |
|---|---|---|---|---|---|---|---|---|---|---|---|---|
| | All Days | M-F and Saturday | M-F and Sunday | M-F only | All Days | M-F and Saturday | M-F and Sunday | M-F Only | All Days | M-F and Saturday | M-F and Sunday | M-F Only |
| **Accessioning / Recording Specimen Receipt** | 3/6 | 3/6 | 0/6 | 0/6 | 0/12 | 12/12 | 0/12 | 0/12 | 0/6 | 0/6 | 0/6 | 6/6 |
| **Courier Operations** | 3/6 | 2/6 | 0/6 | 1/6 | 3/12 | 4/12 | 5/12 | 0/12 | 1/5 | 2/5 | 0/5 | 2/5 |
| **Demographic Data Entry** | 3/6 | 3/6 | 0/6 | 0/6 | 0/12 | 8/12 | 4/12 | 0/12 | 0/6 | 0/6 | 0/6 | 6/6 |
| **Receiving Specimens** | 3/6 | 3/6 | 0/6 | 0/6 | 0/12 | 11/12 | 1/12 | 0/12 | 1/6 | 2/6 | 0/6 | 3/6 |
| **Molecular Testing** | 1/6 | 1/6 | 2/6 | 2/6 | 0/12 | 5/12 | 0/12 | 7/12 | 0/4 | 0/4 | 0/4 | 4/4 |
| **Reporting Non-Time-Critical Results** | 4/6 | 1/6 | 0/6 | 1/6 | 0/11 | 3/11 | 0/11 | 8/11 | 0/6 | 0/6 | 0/6 | 6/6 |
| **Reporting Time- Critical Results** | 5/6 | 1/6 | 0/6 | 0/6 | 2/12 | 9/12 | 0/12 | 1/12 | 0/6 | 1/6 | 0/6 | 6/6 |

*Not all labs reported activities for all categories; total numbers reported are reflected

internal decision may impact the timelienss of individual reporting, but no links to clinical outcomes can be made.

The percentage of specimens reported out within the ACHDNC recommended benchmarks were improved in laboratories with seven days operations compared to those with five or six-day operations (Fig 4). The median percent of programs with specimen results reported out for time-critical disorders within five days of life was greater than 80% in all years for laboratories open seven days a week, while the median in laboratories open six days a week did not reach 65%, and in those open five days a week the median failed to reach 50% of specimens.

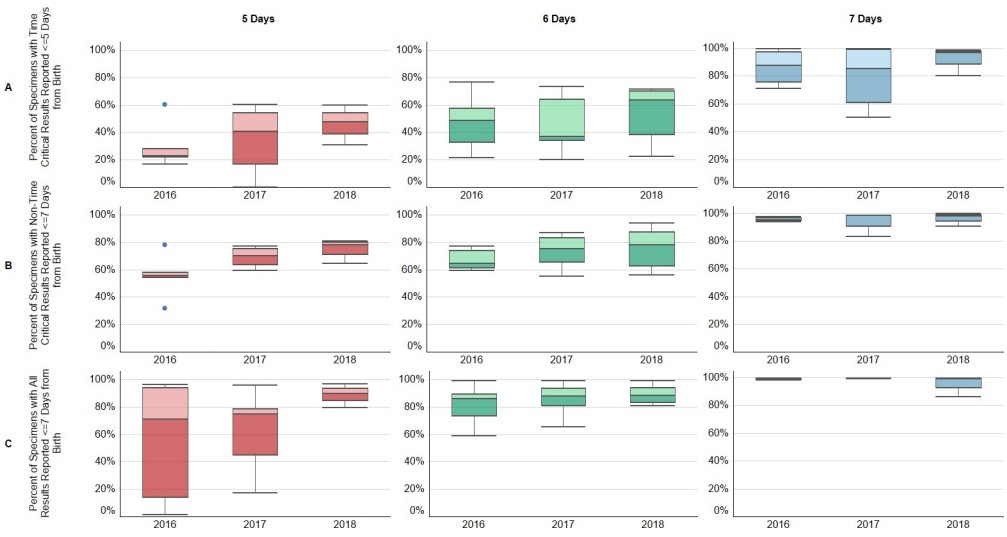

**Fig 4. Newborn screening programs that are open seven days report a higher percentage of results to medical providers in the recommended time period.** Box and whisker plots display the median of the percent of specimens that met the benchmark for each program, with upper and lower quartiles, and range. Panel A: Percent of specimens with results reported out for time-critical results within five days of life at programs open five days (left), six days a week (middle) and seven days a week (right). Panel B: Percent of specimens with results reported for non-time-critical results within seven days of life at programs open five days (left), six days a week (middle) and seven days a week (right). Panel C: Percent of specimens with all results reported within seven days of life at programs open five days (left), six days a week (middle) and seven days a week (right).

The median percent of programs with specimen results reported in a timely manner increased for all laboratories regardless of operating days for both non-time-critical disorders and all results in seven days; however, laboratories open seven days reported results earlier than those open six or five days.

Most NBS programs participating in NewSTEPs 360 have a laboratory housed within their state public health department (n = 18, recorded in 2018), while others contract with external laboratories (n = 7, recorded in 2018). The external laboratory may be managed by a private or commercial entity, or it may be housed within another state public health department. The percent of specimens with results reported within the recommended time frames from birth for both individual state laboratories and external laboratories for 2016–2018 demonstrate improvement for time-critical, non-time-critical, and all result reporting (Fig 5) suggesting that both state and private labs can achieve the same success in newborn screening timeliness.

## Discussion

The overarching ACHDNC timeliness goals are designed to achieve the earliest diagnosis and intervention for infants with time-critical and non-time-critical disorders identified through newborn screening. [7] The introduction of national timeliness goals, paired with a continuous quality improvement program has led to improved times in reporting results to the clinical community, and earlier intervention of affected infants for NBS programs which participated in NewSTEPs 360. NewSTEPs 360 has demonstrated that NBS programs can make progress toward reaching these goals on a population level in a relatively short time through an organized, focused quality improvement effort tailored to the needs of individual states; however, there is room for system improvement.

### State efforts to improve timeliness

During NewSTEPs 360 participation, state programs improved the percentage of specimens collected within ACHDNC's recommended collection time of before 48 hours of life, the percentage of specimens received within two days of collection, and the percentage of results reported out by recommended guidelines. Programs achieved this through different approaches, including (1) implementing educational campaigns with birthing facilities, (2) increasing laboratory hours of operation and workforce schedules, (3) expanding courier service to deliver specimens to the NBS laboratory, (4) changes in laboratory testing methods, (5) using electronic ordering and results reporting with birthing centers, and (6) changes in regulations to require earlier collection. However, the trajectory of improvement and percent improvement varied among participants.

The timely collection of a newborn screening specimen at the birthing facility allows for earlier analysis and reporting. Regulations in three participating states were changed to reflect the shorter national guidelines of 24–48 hours for collection, and remarkable improvements were seen in those states. Additionally, participating NewSTEPs 360 programs developed educational materials, videos, online modules and in-person training sessions to ensure the staff collecting the specimens were knowledgeable about the importance of proper and timely collection and shipping.

NBS laboratories have historically operated during normal working hours on weekdays. However, the increased urgency of many of the new disorders added to the newborn screening panel has changed the paradigm.[4] Many NBS laboratories have shifted their work days to include Saturdays and/or Sundays and extended or modified operating hours throughout the week. Programs within NewSTEPs 360 pursued changes in operating hours, seeking additional funding, increased fees, and modified work schedules for employees. Continued efforts to

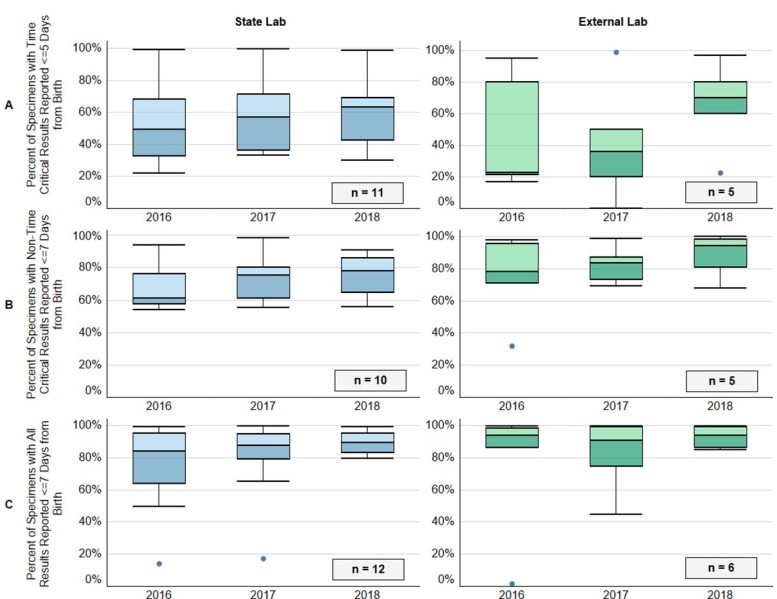

**Fig 5. Newborn screening programs utilizing external laboratories report a higher percentage of results to medical providers in the recommended time period.** Box and whisker plots display the median of the percent of specimens that met the benchmark for each program, with upper and lower quartiles, and range. Panel A: Percent of specimens with results reported out for time-critical results within five days of birth for state labs (left) and external labs (right); Panel B: Percent of specimens with non-time-critical results reported out within seven days of birth for state labs (left) and external labs (right); Panel C: Percent of specimens with all results reported out within seven days after birth for state labs (left) and external labs (right).

increase operating hours so that babies receive the same services independent of the day-of-the-week they were born will decrease the risk of tragic outcomes for individual families. [12]

Most NBS programs are still working to achieve the goal of specimens being delivered to the newborn screening testing laboratory within one or two calendar days of collection. This has been accomplished by individual states through improved shipping from birthing centers, expanded courier systems, increased communication with the couriers, and increased laboratory operating hours to accept specimens. Based upon the largest gaps in timeliness identified in NewSTEPs 360, the best potential for timeliness gains includes increasing the number of days that laboratories are open, adding weekend and holiday couriers, and improving courier services for the transportation of specimens from birthing facilities to newborn screening laboratories.

Improvements in laboratory processes internal to the program were implemented across participating programs with the goal of improving timeliness. For example, one program identified a delay in reporting due to the timing of hemoglobinopathy results, which delayed the reporting of all results. The program changed their incubation and workflow process so that all results could be reported in a timely manner. In another program, a concerted effort was placed on improving the demographic data entry from the dried blood spot cards to improve timely data acquisition and reporting.

Health Information Technology (HIT) solutions hold promise for continuing to improve newborn screening timeliness. Many programs have instituted electronic solutions for data sharing, including electronic orders to improve demographic data transfer, electronic transfer of data, result reporting, and electronic faxing of results. One program implemented electronic ordering of dried blood spot tests, decreasing the time to verify information and initiate testing, initially in four hospitals, then more broadly across the state. The Building Blocks guide

provides guidance to NBS programs to implement HIT solutions that can improve timely orders and reporting of results (S3 File).

## Increased data entry into repository through utilization

One change that was seen as part of NewSTEPs 360 was an increase in data entry in the NewSTEPs data repository. As part of NewSTEPs 360, the repository was configured so that participating states could enter Quality Indicator data monthly vs just yearly. Further, CQI coaches encouraged monthly data entry and tracking. Options to upload data to ease manual entry were also provided, including direct upload of comma-separated-values files (.CSV), and direct assistance with data manipulation within states. NewSTEPs 360 participants utilized real-time data analysis in partnership with their quality improvement coach. The NewSTEPs Data Repository and infographics have been utilized by NBS programs to advocate for additional resources at the local level. For example, programs shared the NewSTEPs 360 data infographics with program leadership to demonstrate the improvements in timeliness metrics that were gained from adding operating hours or couriers, including weekend/holiday couriers. Conversely, other programs were able to demonstrate that they lagged behind the other participating programs and identified resource needs that could help to improve outcomes.

### Unintended consequences

While improving timeliness in newborn screening was the ultimate goal of the NewSTEPs 360 program, timeliness efforts may have unintended consequences. Analytic cutoffs have typically been developed based upon age-based normal ranges for infants who are 24–48 hours of life, and testing infants earlier may impact the accuracy of the tests. Decreasing the accuracy of the screen may result in a high number of specimens flagged for follow-up testing, more infants sent for diagnostic testing, and an increase in false negatives, although this was not reported in this study. Further, earlier screening may limit the time to educate parents in the hospital prior to collecting the screen.

NBS programs have limited resources. Additional demands on staff to meet timeliness goals can limit the time that programs have to improve other program outcomes. Further, changes in programmatic and individual performance expectations may impact staff morale, which in turn affects staff retention. State programs may need to advocate for additional resources to meet timeliness goals, and the other requirements of the NBS program to meet the needs of its newborn population and provide the best outcomes for newborns with a disorder identified by NBS.

### Limitations

The results of this analysis are limited to NewSTEPs 360 funded programs, yet most NBS programs are engaged in activities to achieve ACHDNC timeliness goals and beyond. States NBS programs applied to participate in the NewSTEPs 360 program, potentially introducing a selection bias as they may not be a representative sample of all NBS programs Additionally, NBS program variation in NBS data collection may limit interpretation of QI timeliness data. For instance, the ACHDNC timeliness goals apply to first specimens collected, but some programs were unable to differentiate between first and subsequent specimens, which can result in longer reported timeframes than programs reporting data for first specimens only. Some programs also complete second-tier testing to improve the specificity of the screen, but potentially delaying the final result reporting. In addition, not all programs were able to collect the necessary time stamp of specimen receipt at the laboratory electronically, resulting in limited data reporting for some of the outcomes. There is also significant variability in the definitions

of required data elements on the dried blood spot card, making interpretation across programs difficult. Finally, only a subset of programs provided case level data, limiting the generalizability of the results.

## Conclusions

Newborn screening is one of the most successful public health programs in the US.[13] While states have clear authority with regard to NBS program oversight and monitoring, there is a federal role in supporting states in the implementation of the various components of the newborn screening system and ensuring timely diagnosis and management. The ACHDNC, public health departments, clinical specialists, birthing facilities, midwives, primary care providers, and parents have partnered to improve the newborn screening system. Improving timeliness of reporting of results has been a critical focus.

The individualized approach within NewSTEPs 360 allowed coaches to customize the support provided to the state newborn screening program and, whenever possible, connect one program with another who had shown success in an area. We believe that this structure strengthened the effectiveness of the program. In addition, the NewSTEPs Data Repository played a key role in the success of programs because it allowed (1) participating programs and CQI coaches to identify areas of needs, (2) the NewSTEPs 360 leadership to identify and meet educational needs of the larger group, and (3) the newborn screening community to see the gains made in timeliness.

The achievements of the NBS programs participating in this continuous quality improvement project in partnership with NewSTEPs 360 are the result of the ongoing support by the broader newborn screening community and its commitment to the newborns it serves. Continued success will depend upon that network of support.

## Supporting information

**S1 File.**
(PDF)

**S2 File.**
(PDF)

**S3 File.**
(PDF)

## Acknowledgments

**Disclaimer:** The views expressed in this publication are solely the opinions of the authors and do not necessarily reflect the official policies of the U.S. Department of Health and Human Services or the Health Resources and Services Administration, nor does mention of the department or agency names imply endorsement by the U.S. Government.

## Author Contributions

**Conceptualization:** Marci K. Sontag, Joshua I. Miller, Sarah McKasson, Careema Yusuf, Sikha Singh, Jelili Ojodu, Yvonne Kellar-Guenther.

**Data curation:** Marci K. Sontag, Joshua I. Miller, Sarah McKasson, Ruthanne Sheller, Sari Edelman, Careema Yusuf, Yvonne Kellar-Guenther.

**Formal analysis:** Marci K. Sontag, Joshua I. Miller, Sarah McKasson, Yvonne Kellar-Guenther.

**Funding acquisition:** Marci K. Sontag, Joshua I. Miller, Careema Yusuf, Sikha Singh, Jelili Ojodu, Yvonne Kellar-Guenther.

**Investigation:** Marci K. Sontag, Joshua I. Miller, Sarah McKasson, Ruthanne Sheller, Sari Edelman, Yvonne Kellar-Guenther.

**Methodology:** Marci K. Sontag, Joshua I. Miller, Sarah McKasson, Yvonne Kellar-Guenther.

**Project administration:** Marci K. Sontag, Joshua I. Miller, Sarah McKasson, Jelili Ojodu, Yvonne Kellar-Guenther.

**Resources:** Careema Yusuf, Sikha Singh, Deboshree Sarkar, Joseph Bocchini, Joan Scott, Jelili Ojodu.

**Visualization:** Marci K. Sontag, Joshua I. Miller.

**Writing – original draft:** Marci K. Sontag, Joshua I. Miller, Sarah McKasson, Yvonne Kellar-Guenther.

**Writing – review & editing:** Marci K. Sontag, Sarah McKasson, Ruthanne Sheller, Sari Edelman, Careema Yusuf, Sikha Singh, Deboshree Sarkar, Joseph Bocchini, Joan Scott, Jelili Ojodu.

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
