## [Decision Letter · Decision Letter 0]

17 Nov 2019

PONE-D-19-27133

Newborn screening timeliness quality improvement initiative: Impact of national recommendations and data repository

PLOS ONE

Dear Dr Marci K. Sontag and co-workers,

Thank you for submitting your manuscript to PLOS ONE. After careful consideration, we feel that it has merit but does not fully meet PLOS ONE’s publication criteria as it currently stands. Therefore, we invite you to submit a revised version of the manuscript that addresses the points raised during the review process.

Some minor issues like the abstract's structure, shortening by deleting repetitions and some context for non-Americans. 

We would appreciate receiving your revised manuscript by Dec 23 2019 11:59PM. To enhance the reproducibility of your results, we recommend that if applicable you deposit your laboratory protocols in protocols.io, where a protocol can be assigned its own identifier (DOI) such that it can be cited independently in the future. For instructions see: http://journals.plos.org/plosone/s/submission-guidelines#loc-laboratory-protocols

We look forward to receiving your revised manuscript.

Kind regards,

Jacobus P. van Wouwe, MD PhD

Academic Editor

PLOS ONE

Journal Requirements:

2. For studies reporting research involving human participants, PLOS ONE requires authors to confirm that this specific study was reviewed and approved by an institutional review board (ethics committee) before

the study began. Please provide the specific name of the ethics committee/IRB that approved your study, or explain why you did not seek approval in this case.

4.  Thank you for stating the following in the Financial Disclosure section:

The project described in this article was funded by the Department of Health and Human Services, Health Resources and Services administration under Cooperative Agreement # UG8MC28554. (MKS, JIM, SM, CY, RS, SE, SS, JO)

The funders played a role in decision to publish and Preparation of the manuscript.

We note that one or more of the authors are employed by a commercial company: CI International.

Additional Editor Comments:

The reviewers gave some practical suggestions how to improve your well written manuscript on this relevant subject. The structure of your abstract, shortening the discussion by deleting repeats, and the suggestion to add some context for the international community will further improve your manuscript. The examples the reviewers have pointed out seem easy to answer and I do hope to see your revision being submitted soon! This manuscript needs to get published! Congrats.

Reviewers' comments:

Reviewer's Responses to Questions

**Comments to the Author**

1. Is the manuscript technically sound, and do the data support the conclusions?

Reviewer #1: Yes

Reviewer #2: Partly

Reviewer #3: Yes

2. Has the statistical analysis been performed appropriately and rigorously? 

Reviewer #1: Yes

Reviewer #2: I Don't Know

Reviewer #3: Yes

3. Have the authors made all data underlying the findings in their manuscript fully available?

Reviewer #1: Yes

Reviewer #2: Yes

Reviewer #3: Yes

4. Is the manuscript presented in an intelligible fashion and written in standard English?

Reviewer #1: Yes

Reviewer #2: Yes

Reviewer #3: Yes

5. Review Comments to the Author

Reviewer #1: Newborn screening timeliness quality improvement initiative: Impact of national recommendations and data repository

This paper provides an overview of quality improvement initiatives taken on by state newborn screening programs to meet timeliness goals set by theSecretary of Health and Human Services’ Advisory Committee on Heritable Disorders in Newborns and Children. Using a data repository, the outcomes of these initiatives could be tracked and monitored between an individual state and the aggregate of all participating screening programs.

Thank you for giving me the opportunity to review this paper. Overall, I am favorable for the publication of this important summary of state efforts to meet national recommendations. Below are my comments and suggestions for edits within each section of the paper.

Abstract:

Lines 67-69: Consider starting the abstract with the goal of newborn screening “Early identification and treatment of affected infants prior to the onset of symptoms”. Then define the necessary and time critical steps to accomplish the goal (ie…specimen collection, transport, testing, and reporting). The words “newborn screening activities” make it seem optional or less significant. These are steps are critically important to accomplishing the mission.

Lines 77-78: Give a bit more detail on the necessary steps (ie…specimen collection, transport, testing, and reporting). This will help lead into the results summary in line 79-90.

Line 91: This is the first time you mention NewSTEPs 360. Please define NewSTEPs 360, perhaps in the background or the methods section of the abstract.

Introduction:

Line 101: Did you want to mention that clinics also collect newborn screening specimens, in addition to hospitals and midwives?

Line 104: Consider rewording to “medical personnel to facilitate additional testing to confirm or exclude a diagnosis and also initiate the required appropriate intervention.” I see facilitation of additional testing as being the primary goal (first step). Intervention (although absolutely important under certain conditions) may or may not be needed depending upon the results.

Figure 1: I struggled with figure 1. Your figure legend talks about NewSTEPs 360 and then lists five categories of activities performed by states to improve timeliness, but your figure only shows the process of newborn screening. I would suggest removing this text from the legend OR adding details to the figure to describe the five categories.

Line 171-177: Consider rewording the last paragraph a bit to reflect each pieces of your review. You “summarized” the quality improvement initiatives. You “evaluated” the timeliness data for initial specimen collection, delivery from the birthing center, etc, over a certain time period. You “correlated” or “analyzed” or “assessed” the impact of individual program activities to improve timeliness.

Methods:

Lines 193-203: Consider rewording. State first what is available to the public, followed by what is available to registered users and then finally what states have access to after providing the MOU. The paragraph is a bit confusing.

Line 210-224 and table 3: Did you want to use the benchmark of two calendar days for the timing between “specimen collection to receipt at lab”? Or only one day in transit, as shown in table 3?

Line 210: You mentioned in line 210 that you have added two additional timeliness benchmarks, what are the two benchmarks (2 days in transit and time to reporting?), but one is a modification to a metric and the other is a new metric? Please consider rewording the paragraph to provide clarity.

Line 247: You refer to PDSA as a personality tool. I really thought it was a tool to assess a change that is implemented. The word personality confused me as well as how this tool could support team growth and team member’s roles.

Results:

Table 5: Define the “N” as the number of programs that provided data for a give category. Maybe in a footnote?

Table 5: Please clearly differentiate between reporting of results “presumptive positive for time critical disorders” “presumptive positive for non-time critical disorders”, and reporting of “all (normal and presumptive positive) screening results” OR reporting of “all normal results”?

Line 323: I just want to make sure the difference between each of the three categories (time critical/non-time critical/all) is clearly communicated. Plus…maybe this in inherently obvious…but I think it is necessary to state again…that the reporting of presumptive positive NBS results doesn’t mean that the patient is affected with the disease.

Figure 2: Image is not clear/fuzzy. And it could also be larger. But maybe this will be addressed during the formatting process.

Line 344: “did not demonstrate change” OR “did demonstrate change”. It appears that the percentage when up slightly…but maybe this is not a significant change?

Table 7: This table is a bit confusing. For the six laboratories open 7 days a week, it is correct to read that 3 laboratories accession specimens all seven days and 3 laboratories accession specimens ONLY on Saturday? For the laboratory only open Monday through Friday, why do they have activities such as receiving specimens on Saturday? I believe I know what you mean to convey…but it is not inherently clear.

Line 407-408: The last sentence regarding the overall median for all results reported measures, comparing external versus state lab, is hard to appreciate from figure 5. It appears that the median would be closer to 80% for external labs. Maybe I would remove this sentence or clarify.

Discussion:

General comment: I really appreciate how you linked specific programmatic activities to the potential improvement seen within certain metrics.

Line 521: Include within the limitations section of the discussion how second tier (or even third tier) testing (molecular or biochem) performed by laboratories may impact the time to reporting, even for critical conditions.

Reviewer #2: Thank you for this interesting work. As time is of essence for many neonatal screening conditions, this work is very relevant for the stakeholders involved.

As suggestions I would like to put forward that I think the manuscript might be in use of extra context. The relevance of the topic is undisputed, but if the benchmark and the international context is presented more clearly, the reader would be able to grasp the relevance better. For example:

- how many cases are missed due to lack of timeliness? Which is hard to make concrete, because it would also require post-mortem genetic testing?

- what was the starting point of timeliness; there is mentioning of increasing time-critical disorders, but is this due to the type of disorders added to the RUSP or the revision of what a time-critical disorder is?

- how is this embedded or does it compare to international systems? Not all countries have birthing facilities / different facilities / deliveries at home, and the time of collection also differs quite a lot between countries.

- was an implementation theory or model used? How did all parts fit into that?

- Why was the selection based on a competative application process?

- How is the quality and standardization of data in the repository ensured?

Furthermore, I think the manuscript could be quite a bit shorter on the results if the tables are made more legible. Especially tables 5 and 6 are hard to read, because as a reader you are not always sure what you are looking at. Especially combined with the figures and the text, it seems there is a lot of double information, but the sources (how many labs answered which questions, i.e. how was dealt with missing values) are not always clear while potential stratification and/or combination of the variables is not made. Perhaps some information is better to put into tables instead of figures?

Taking the above into account, I think it is important and interesting work, but to be an international scientific publication it seems to be in need of a better handle on highlighting only or more of the really interesting results to the reader; what tables and figures are really relevant?, and better embedding in international scientific literature, unless it is more or less solely aimed at US readers, but still evidence-based implementation theories / model could be applied.

Reviewer #3: Dear authors,

I have written this manuscript with pleasure and I feel it is suitable for publication in PLOSONE.

This is a well-written paper on the continuous improvement of the primary process of newborn screening. It is a showcase of how data concerning this process should be handled and how they can be used to monitor continuous improvement, providing the necessary elements of a PDCA cycle. While scientifically, the data may not be top-notch, for the field of neonatal screening it is an example, also eliciting the ‘machinery’, in terms of centralized databases and secure access, and predefined criteria and definitions, that is needed to make this work.

You make the necessary selections (in terms of handling cards without demographic data or cards that were flagged for recollection) to come to sensible outcome measures.

I have very few issues with this paper.

Line 180

Particpants in Newsteps. Selecting states is not selecting others. Can you explain what states were not selected, why, and whether this could introduce selection bias? (e.g. at limitations of study).

Line 407

overall median was observed. Median of what?

Line 491 -500.

I would like to know whether at the site of the laboratory, these CQI reports were generated automatically, querying the databases of the individual laboratories.. This information is important as it superior to manual transfer of data. If the data is entered manually in the files to upload in the repository, is it curated or checked at the site of the repository? This also applies on ln 204-section Quality Indicator data.

The discussion is quite lengthy and contains repeats of what was already said in the reminder of the paper. Moreover, the discussion is used to describe developments within the programs (e.g. a lengthy description of the efforts made to aIso work on weekend days feel that the authors can be more concise and ( Lines 512-520This is true, but does not relate to the study results.) . I feel the Conclusions section is a double of what has been said before in the paper, especially the discussion.

6. PLOS authors have the option to publish the peer review history of their article (what does this mean?). If published, this will include your full peer review and any attached files.

Reviewer #1: No

Reviewer #2: No

Reviewer #3: No

---

## [Author Response · Author response to Decision Letter 0]

24 Jan 2020

Specific Comments from Reviewer #1: 

Abstract:

• Lines 67-69: Consider starting the abstract with the goal of newborn screening “Early identification and treatment of affected infants prior to the onset of symptoms”. Then define the necessary and time critical steps to accomplish the goal (ie…specimen collection, transport, testing, and reporting). The words “newborn screening activities” make it seem optional or less significant. These are steps are critically important to accomplishing the mission.

AND 

Lines 77-78: Give a bit more detail on the necessary steps (ie…specimen collection, transport, testing, and reporting). This will help lead into the results summary in line 79-90.

Response: Thank you for this comment. We have adjusted the abstract to reflect these changes in both the background and methods section of the abstract. 

• Line 91: This is the first time you mention NewSTEPs 360. Please define NewSTEPs 360, perhaps in the background or the methods section of the abstract.

Response: We have added NewSTEPs 360 in both the background and methods section of the abstract.

Introduction:

• Line 101: Did you want to mention that clinics also collect newborn screening specimens, in addition to hospitals and midwives?

Response: The reviewer is correct. While most of the focus of the newborn screening timeliness activities were on the first specimen, clinics do indeed collect specimens. We have modified the introduction to reflect this. 

• Line 104: Consider rewording to “medical personnel to facilitate additional testing to confirm or exclude a diagnosis and also initiate the required appropriate intervention.” I see facilitation of additional testing as being the primary goal (first step). Intervention (although absolutely important under certain conditions) may or may not be needed depending upon the results.

Response: The reviewer brings up an excellent point. We believe the initiation of an appropriate intervention to be the ultimate goal for affected infants, and many times a confirmatory diagnosis may come months after an intervention starts. The reviewer accurately notes that classification of infants into affected vs unaffected is critically important. We have rephrased this sentence as suggested, to indicate diagnosis should occur first. 

• Figure 1: I struggled with figure 1. Your figure legend talks about NewSTEPs 360 and then lists five categories of activities performed by states to improve timeliness, but your figure only shows the process of newborn screening. I would suggest removing this text from the legend OR adding details to the figure to describe the five categories.

Response: The categories of activities are displayed in the linked boxes under each step in the NBS process. We believe figure one more clearly describes the process and the supporting activities within the process that were implemented within NewSTEPs 360. 

• Line 171-177: Consider rewording the last paragraph a bit to reflect each pieces of your review. You “summarized” the quality improvement initiatives. You “evaluated” the timeliness data for initial specimen collection, delivery from the birthing center, etc, over a certain time period. You “correlated” or “analyzed” or “assessed” the impact of individual program activities to improve timeliness.

Response: Thank you, we have made the edits as suggested by the reviewer.

Methods:

• Lines 193-203: Consider rewording. State first what is available to the public, followed by what is available to registered users and then finally what states have access to after providing the MOU. The paragraph is a bit confusing.

Response: This paragraph has been modified to better describe the processes in place for data sharing and data access. 

• Line 210-224 and table 3: Did you want to use the benchmark of two calendar days for the timing between “specimen collection to receipt at lab”? Or only one day in transit, as shown in table 3?

Response: This is a great observation. To avoid additional complicated descriptions we have chosen to present one day transit within column C of Table 3. We could have also modified the birth to specimen collection window to be one day, rather than two days. Each of these benchmarks is a bit arbitrary. We wanted to derive a metric to allow NBS programs to assess their performance in laboratory processes, and we did so using the metrics already defined by the ACHDNC, resulting in two days of laboratory testing for time-critical and four days for non-time-critical. 

• Line 210: You mentioned in line 210 that you have added two additional timeliness benchmarks, what are the two benchmarks (2 days in transit and time to reporting?), but one is a modification to a metric and the other is a new metric? Please consider rewording the paragraph to provide clarity.

Response: This is an excellent point. We edited the paragraph to reflect that we modified one benchmark and added a new one. 

• Line 247: You refer to PDSA as a personality tool. I really thought it was a tool to assess a change that is implemented. The word personality confused me as well as how this tool could support team growth and team member’s roles.

Response: The PDSA tool that we have developed is a tool that does assess personalities, based on the quality improvement technique that you refer to. We have added a sentence that we believe clarifies the use of the tool.

Results:

• Table 5: Define the “N” as the number of programs that provided data for a give category. Maybe in a footnote?

Response: Excellent point. We have added a note to the table legend to reflect the N is the number of programs that provided data for the quality indicator. 

• Table 5: Please clearly differentiate between reporting of results “presumptive positive for time critical disorders” “presumptive positive for non-time critical disorders”, and reporting of “all (normal and presumptive positive) screening results” OR reporting of “all normal results”?

Response: Thank you for noting this, we have made the changes as suggested.

• Line 323: I just want to make sure the difference between each of the three categories (time critical/non-time critical/all) is clearly communicated. Plus…maybe this in inherently obvious…but I think it is necessary to state again…that the reporting of presumptive positive NBS results doesn’t mean that the patient is affected with the disease.

Response: The definitions of time-critical, non-time-critical are provided in the background of the document. We have added a sentence to the legend in Table 5 to define presumptive positive results. 

• Figure 2: Image is not clear/fuzzy. And it could also be larger. But maybe this will be addressed during the formatting process.

Response: Thank you, we will confirm that the figure is appropriately formatted upon final submission. 

• Line 344: “did not demonstrate change” OR “did demonstrate change”. It appears that the percentage when up slightly…but maybe this is not a significant change?

Response: The reviewer is correct, the program median for reporting was not significantly different (88.9 to 89.5% across the 3 years). 

• Table 7: This table is a bit confusing. For the six laboratories open 7 days a week, it is correct to read that 3 laboratories accession specimens all seven days and 3 laboratories accession specimens ONLY on Saturday? For the laboratory only open Monday through Friday, why do they have activities such as receiving specimens on Saturday? I believe I know what you mean to convey…but it is not inherently clear.

Response: Thank you for this comment. We have revised the table to make is less confusing. Specifically:

- We added M-F to the columns that indicate that testing occurs on Saturday or Sunday (M-F and Saturday, M-F and Sunday). 

- We revised the table to reflect that the number of days a laboratory is open is defined by the number of days the laboratory tests specimens.

• Line 407-408: The last sentence regarding the overall median for all results reported measures, comparing external versus state lab, is hard to appreciate from figure 5. It appears that the median would be closer to 80% for external labs. Maybe I would remove this sentence or clarify.

Response: This sentence was removed as suggested.

Discussion:

• General comment: I really appreciate how you linked specific programmatic activities to the potential improvement seen within certain metrics.

Response: Thank you! 

• Line 521: Include within the limitations section of the discussion how second tier (or even third tier) testing (molecular or biochem) performed by laboratories may impact the time to reporting, even for critical conditions.

Response: Thank you for this comment. This is an important word of caution for newborn screening programs, we have added it to the limitation section. 

Specific Comments from Reviewer #2: 

As suggestions I would like to put forward that I think the manuscript might be in use of extra context. The relevance of the topic is undisputed, but if the benchmark and the international context is presented more clearly, the reader would be able to grasp the relevance better. For example:

How many cases are missed due to lack of timeliness? Which is hard to make concrete, because it would also require post-mortem genetic testing? 

Response: There have been some reports of tragic cases in which an infant has died or had poor outcomes due to a delay in newborn screening reporting, however it is hard to really assess this in a systematic way.

What was the starting point of timeliness; there is mentioning of increasing time-critical disorders, but is this due to the type of disorders added to the RUSP or the revision of what a time-critical disorder is?

Response: We added more details to the section regarding the expansion of the panel of NBS disorders (in the Introduction, Section on Timeliness in Newborn Screening) to reflect that the change really occurred with the introduction of MS/MS)

How is this embedded or does it compare to international systems? Not all countries have birthing facilities / different facilities / deliveries at home, and the time of collection also differs quite a lot between countries.

Response: This manuscript is definitely focused on the US systems. Many other countries do not screen for disorders using MS/MS and therefore they do not have the same urgency for early detection. Further, as the reviewer points out, other countries may have more babies born outside of traditional birthing facilities, and may also have newborn screen specimens collected at different times, coinciding with a home-visit from a nurse. These differences are beyond the scope of this manuscript. 

Was an implementation theory or model used? How did all parts fit into that?

Response: No implementation theory or model was used but state activities addressed structural characteristics and the NewSTEPs 360 team provided ongoing implementation support activities (e.g. coaching) and building CQI self-efficacy for the NBS team

Why was the selection based on a competitive application process?

Response: Participation was based on a competitive process because each program had to demonstrate that they were prepared to make changes in their program and collect data to monitor progress.

How is the quality and standardization of data in the repository ensured?

Response: The data entered into the NewSTEPs repository are checked via data validation on each field upon data entry, and data inspection by the NewSTEPs staff, and NBS programs are queried to clarify identified outliers or suspected errors. 

• Furthermore, I think the manuscript could be quite a bit shorter on the results if the tables are made more legible. Especially tables 5 and 6 are hard to read, because as a reader you are not always sure what you are looking at. Especially combined with the figures and the text, it seems there is a lot of double information, but the sources (how many labs answered which questions, i.e. how was dealt with missing values) are not always clear while potential stratification and/or combination of the variables is not made. Perhaps some information is better to put into tables instead of figures?

Response: We have edited tables 5 and 6 to be clearer. We believe the stratification of the data into the tables and figures is important as there are nuanes in these data that are important in the interpretation of the results of the study. 

• Taking the above into account, I think it is important and interesting work, but to be an international scientific publication it seems to be in need of a better handle on highlighting only or more of the really interesting results to the reader; what tables and figures are really relevant?, and better embedding in international scientific literature, unless it is more or less solely aimed at US readers, but still evidence-based implementation theories / model could be applied.

Response: The ACHDNC guidelines are intended for U.S. newborn screening programs, and many of the processes that are described in the manuscript are unique to the U.S.; however, we believe that many lessons can be gleaned for international audiences. We have better described the tables and figures and we believe that the figures are necessary for the reader to understand the nuances of NBS programs in the U.S. 

Specific Comments from Reviewer #3:

• Line 180: Participants in Newsteps. Selecting states is not selecting others. Can you explain what states were not selected, why, and whether this could introduce selection bias? (e.g. at limitations of study).

Response: We have added a sentence to the limitations section to indicate that the application to NewSTEPs 360 could introduce a selection bias. 

• Line 407: overall median was observed. Median of what?

Response: This sentence was deleted in response to comment from Reviewer 1. 

• Line 491 -500.

I would like to know whether at the site of the laboratory, these CQI reports were generated automatically, querying the databases of the individual laboratories.. This information is important as it superior to manual transfer of data. If the data is entered manually in the files to upload in the repository, is it curated or checked at the site of the repository? This also applies on ln 204-section Quality Indicator data.

Response: The reviewer has identified a critical point in the sharing of electronic data. Unfortunately, there are no methods available to automatically query the databases of all of the individual laboratories. The NewSTEPs team has made progress within some states and some of the Laboratory Information Management Systems (LIMS) vendors have individual solutions, this is not available broadly. Data checks are also implemented for data uploaded using automated upload tools. 

• The discussion is quite lengthy and contains repeats of what was already said in the reminder of the paper. Moreover, the discussion is used to describe developments within the programs (e.g. a lengthy description of the efforts made to aIso work on weekend days feel that the authors can be more concise and ( Lines 512-520This is true, but does not relate to the study results.) . I feel the Conclusions section is a double of what has been said before in the paper, especially the discussion.

Response: Thank you for noting this. We have shortened both the conclusion and discussion section in light of your suggestions and we believe it reads better in its more concise form.

---

## [Decision Letter · Decision Letter 1]

7 Feb 2020

PONE-D-19-27133R1

Newborn screening timeliness quality improvement initiative: Impact of national recommendations and data repository

PLOS ONE

Dear Dr Marci K Sontag and co-authors,

Thank you for submitting your manuscript to PLOS ONE. After careful consideration, we feel that it has merit but does not fully meet PLOS ONE’s publication criteria as it currently stands. Therefore, we invite you to submit a revised version of the manuscript that addresses the points raised during the review process and make the clarifications as asked for.

We would appreciate receiving your revised manuscript by Mar 23 2020 11:59PM. To enhance the reproducibility of your results, we recommend that if applicable you deposit your laboratory protocols in protocols.io, where a protocol can be assigned its own identifier (DOI) such that it can be cited independently in the future. For instructions see: http://journals.plos.org/plosone/s/submission-guidelines#loc-laboratory-protocols

We look forward to receiving your revised manuscript.

Kind regards,

Jacobus P. van Wouwe, MD PhD

Academic Editor

PLOS ONE

Additional Editor Comments (if provided):

Please answer the issues raised by the reviewer to increase the readability of your manucript.

Reviewers' comments:

Reviewer's Responses to Questions

**Comments to the Author**

1. If the authors have adequately addressed your comments raised in a previous round of review and you feel that this manuscript is now acceptable for publication, you may indicate that here to bypass the “Comments to the Author” section, enter your conflict of interest statement in the “Confidential to Editor” section, and submit your "Accept" recommendation.

Reviewer #1: (No Response)

Reviewer #3: All comments have been addressed

2. Is the manuscript technically sound, and do the data support the conclusions?

Reviewer #1: Yes

Reviewer #3: Yes

3. Has the statistical analysis been performed appropriately and rigorously? 

Reviewer #1: Yes

Reviewer #3: Yes

4. Have the authors made all data underlying the findings in their manuscript fully available?

Reviewer #1: Yes

Reviewer #3: Yes

5. Is the manuscript presented in an intelligible fashion and written in standard English?

Reviewer #1: Yes

Reviewer #3: Yes

6. Review Comments to the Author

Reviewer #1: The revised manuscript and remarks from the authors addressed my initial concerns. Below are some additional edits for the authors to consider.

Line 102

o Consider rewording to “in the first week of life”, instead of “first days or weeks of life”. Current recommendations target identification within the first week of life.

Line 113 (figure 1)

o Consider adding numbers to the figure to link the legend. For example, labeling the box within the figure “QI Solution 1”, which would link to the “1)” in the legend.

Line 157

o Consider rewording to “…barriers to timely NBS through technical and financial assistance”. I don’t understand what you mean by “utilization of technology”, unless you are referring to the visualization of performance metrics, quality indicators?

Line 210

o State that role-based access control was determined/granted at the individual NBS program level.

Line 220-224

o Wording is a bit unclear and could be made more concise

Line 268 (table 4)

o I “want” to link the various strategies (education, courier, NBS laboratory, etc) to the QI solutions listed in figure 1. It is possible to reconfigure this table to assign each strategy to the five QI solutions, highlighted in table 1.

Line 310 (table 5)

o I think the year should be “2016” instead of “2015”

Line 313 (figure 2)

o I am not sure if showing the lines on the graph for each individual state (connecting between the years and box plots) adds additional value. It may be more “clean” to only show the box and whisker plots. But this is just a preference…

Line 377-379

o These couple of sentences were confusing. The cases with confirmed diagnosis were identified between 2016-2018, so please consider including 2018 data to lines 377 to 379. Is there a different way to visualize the data, rather than text? Another table perhaps? Or maybe just consider removing, as you have the summary information above in table 6.

Line 394

o Spelling error “operations”

Line 394 (table 7)

o I am still struggling with table 7. For example, it doesn’t make sense that a state would report testing 7 days a week…but only receiving specimen on six days (M-F and Saturday) OR only reporting non-time critical results on Monday through Friday.

o Because it is difficult to make sense of the table and because it is somewhat arbitrary (based upon reported practices of a given state), I think I would remove the table.

o The text can remain as is…except maybe to highlight that each state “reported whether they were open 5, 6, or 7 days” and that “activities performed on a given day, vary between states”.

o If you wanted to keep the table, please add additional sentences to the results or discussion that suggest how various activities performed 5, 6, or 7 days a week may have impacted the timeliness.

Line 405

o What do you mean by “laboratories open seven days demonstrated better outcomes than those open six or five days”? What is the “outcome” by which you used to evaluate whether a state performed better.

Line 420-426 and 430-437

o I don’t think it is necessary to include this information on whether it was a state laboratory or an external laboratory, unless you are trying to make the argument that “people might perceive that one is better than another…but in reality, both state labs and private labs can achieve the same success”? Again, just trying to understand the value of including this section.

Line 451-460

o Again, can we link these six different approaches listed to the 5 different QI solutions? Also see Line 268 (table 4) comment above.

Line 463

o Spacing error in text

Line 520

o Instead of the words “consult parents”, maybe consider using “educate parents”.

Lines 522-528

o Wording is a bit unclear and could be made more concise

Reviewer #3: The authors have skillfully taken care of my comments. I thank them for their work and congratulate them with their work.

7. PLOS authors have the option to publish the peer review history of their article (what does this mean?). If published, this will include your full peer review and any attached files.

Reviewer #1: No

Reviewer #3: No

---

## [Author Response · Author response to Decision Letter 1]

11 Mar 2020

• Line 102

o Consider rewording to “in the first week of life”, instead of “first days or weeks of life”. Current recommendations target identification within the first week of life.

RESPONSE: Changed as suggested 

• Line 113 (figure 1)

o Consider adding numbers to the figure to link the legend. For example, labeling the box within the figure “QI Solution 1”, which would link to the “1)” in the legend.

RESPONSE: We have made the change to number to QI solutions as suggested, here and throughout the manuscript. Thank you for this suggestion.

• Line 157

o Consider rewording to “…barriers to timely NBS through technical and financial assistance”. I don’t understand what you mean by “utilization of technology”, unless you are referring to the visualization of performance metrics, quality indicators?

RESPONSE: Changed as suggested 

• Line 210

o State that role-based access control was determined/granted at the individual NBS program level. 

RESPONSE: Changed as suggested 

• Line 220-224

o Wording is a bit unclear and could be made more concise

Response: Changed as reflected here: Benchmarks were adapted directly from the ACHDNC recommendations. Few newborn screening programs were able to meet the ambitious 24-hours from specimen collection to receipt by laboratory benchmark set by the ACHDNC. In response to this, NewSTEPs created an additional benchmark of two calendar days to assess time elapsed from specimen collection to receipt at the laboratory as an intermediary step.

• Line 268 (table 4)

o I “want” to link the various strategies (education, courier, NBS laboratory, etc) to the QI solutions listed in figure 1. It is possible to reconfigure this table to assign each strategy to the five QI solutions, highlighted in table 1.

Response: This is a great suggestion. We have modified Table 4 to reflect the numbered QI solutions in Figure 1. 

• Line 310 (table 5) 

o I think the year should be “2016” instead of “2015”

Response: You are correct. Thank you for catching this. 

• Line 313 (figure 2)

o I am not sure if showing the lines on the graph for each individual state (connecting between the years and box plots) adds additional value. It may be more “clean” to only show the box and whisker plots. But this is just a preference…

Response: We have left this figure in its initial format as we believe the changes in individual programs is important in quality improvement projects such as this one. 

• Line 377-379

o These couple of sentences were confusing. The cases with confirmed diagnosis were identified between 2016-2018, so please consider including 2018 data to lines 377 to 379. Is there a different way to visualize the data, rather than text? Another table perhaps? Or maybe just consider removing, as you have the summary information above in table 6.

Response: We have removed these sentences as suggested. They do not significantly add to the content. 

• Line 394

o Spelling error “operations”

RESPONSE: Changed as suggested

• Line 394 (table 7)

o I am still struggling with table 7. For example, it doesn’t make sense that a state would report testing 7 days a week…but only receiving specimen on six days (M-F and Saturday) OR only reporting non-time critical results on Monday through Friday. 

o Because it is difficult to make sense of the table and because it is somewhat arbitrary (based upon reported practices of a given state), I think I would remove the table. 

o The text can remain as is…except maybe to highlight that each state “reported whether they were open 5, 6, or 7 days” and that “activities performed on a given day, vary between states”.

o If you wanted to keep the table, please add additional sentences to the results or discussion that suggest how various activities performed 5, 6, or 7 days a week may have impacted the timeliness. 

o 

RESPONSE: We believe this table is an important table in the context of this manuscript. We have added text in the results section to describe some of the nuances of individual programs, and we have made additional edits as suggested by the reviewer to highlight that the laboratory hours are reported by an individual patient and that daily activities vary between states. 

• Line 405

o What do you mean by “laboratories open seven days demonstrated better outcomes than those open six or five days”? What is the “outcome” by which you used to evaluate whether a state performed better.

RESPONSE: The outcome highlighted here is result reporting. We have changed the text to remove the ambiguity. Thank you for identifying this. 

• Line 420-426 and 430-437 - I don’t think it is necessary to include this information on whether it was a state laboratory or an external laboratory, unless you are trying to make the argument that “people might perceive that one is better than another…but in reality, both state labs and private labs can achieve the same success”? Again, just trying to understand the value of including this section.

Response: The reviewer brings up a good point. We have modified this section to indicate that both state and private labs can achieve good timeliness outcomes. We believe that this is important to note as there are a number of stakeholders that would like to suggest the privatization of NBS programs. 

• Line 451-460

o Again, can we link these six different approaches listed to the 5 different QI solutions? Also see Line 268 (table 4) comment above.

Response: We rearranged the numbering and modified the wording to be consistent with the approaches listed in Figure 1. We have not called it out directly, but it is now in parallel which we believe will be more satisfying for the astute reader. 

• Line 463

o Spacing error in text

RESPONSE: Changed as suggested 

• Line 520

o Instead of the words “consult parents”, maybe consider using “educate parents”. 

RESPONSE: Changed as suggested 

• Lines 522-528

o Wording is a bit unclear and could be made more concise

RESPONSE: We modified the paragraph to be more concise.

---

## [Editor Report · Decision Letter 2]

16 Mar 2020

Newborn screening timeliness quality improvement initiative: Impact of national recommendations and data repository

PONE-D-19-27133R2

Dear Dr. Marci K. Sontag and co-workers,

We are pleased to inform you that your manuscript has been judged scientifically suitable for publication and will be formally accepted for publication once it complies with all outstanding technical requirements.

With kind regards,

Jacobus P. van Wouwe, MD PhD

Academic Editor

PLOS ONE

Additional Editor Comments (optional):

Although the line numbers did not match, by the color of the not too many changes it is clear that the reviewers' suggestions have been followed and the manuscript has likewise improved.
---

## [Editor Report · Acceptance letter]

18 Mar 2020

PONE-D-19-27133R2 

Newborn screening timeliness quality improvement initiative: Impact of national recommendations and data repository 

Dear Dr. Sontag:

I am pleased to inform you that your manuscript has been deemed suitable for publication in PLOS ONE. Congratulations! Your manuscript is now with our production department. 

With kind regards,

on behalf of

Dr. Jacobus P. van Wouwe 

Academic Editor

PLOS ONE